# Hexagons all the way down: grid cells as a conformal isometric map of space

**Vemund Sigmundson Schøyen**[1]\*, **Kosio Beshkov**[1☯], **Markus Borud Pettersen**[2☯], **Erik Hermansen**[3], **Konstantin Holzhausen** [4], **Anders Malthe-Sørenssen**[4], **Marianne Fyhn**[1], **Mikkel Elle Lepperød** [2,4]\*

**1** Department of Biosciences, University of Oslo, Oslo, Norway, **2** Department of Physics, University of Oslo, Oslo, Norway, **3** Department of Mathematical Sciences, Norwegian University of Science and Technology, Trondheim, Norway, **4** Department of Numerical Analysis and Scientific Computing, Simula Research Laboratory AS, Oslo, Norway

☯ These authors contributed equally to this work.

\* vemund@live.com (VSS); mikkel@simula.no (MEL)

**Data availability statement:** All relevant data and code for simulations and analysis can be

## Abstract

Grid cells in the entorhinal cortex are known for their hexagonal spatial activity patterns and are thought to provide a neural metric for space, and support path integration. In this study, we further investigate grid cells as a metric of space by optimising them for a conformal isometric (CI) map of space using a model based on a superposition of plane waves. By optimising the phases within a single grid cell module, we find that the module can form a CI of two-dimensional flat space with phases arranging into a regular hexagonal pattern, supporting an accurate spatial metric. Additionally, we find that experimentally recorded grid cells exhibit CI properties, with one example module showing a phase arrangement similar to the hexagonal pattern observed in our model. These findings provide computational and preliminary experimental support for grid cells as a CI-based spatial representation. We also examine other properties that emerge in CI-optimised modules, including consistent energy expenditure across space and the minimal cell count required to support unique representation of space and maximally topologically persistent toroidal population activity. Altogether, our results suggest that grid cells are well-suited to form a CI map, with several beneficial properties arising from this organisation.

## Author summary

Grid cells in the brain's entorhinal cortex are neurons that fire in hexagonal patterns as an animal moves through space, effectively creating an internal map of the environment. These cells are believed to perform two main functions: path integration—the process by which animals determine their position based on self-movement—and serving as a spatial metric, accurately representing distances and angles within their surroundings. In our study, we focus on the latter function: how grid cells act as a metric of space. Using a mathematical model based on the superposition of plane waves, we investigate how and

found at https://github.com/bioAI-Oslo/
CI-grid-cells alongside data within the
manuscript and its Supporting Information
files.

**Funding:** This work was supported by the
Research Council of Norway (300504 to AMS)
https://www.forskningsradet.no. The funders
had no role in study design, data collection and
analysis, decision to publish, or preparation of
the manuscript.

**Competing interests:** The authors have
declared that no competing interests exist.

when grid cells naturally preserve distances and angles in flat two-dimensional space.
Our findings demonstrate that grid cells can inherently form a conformal isometric (CI)
map, maintaining the geometric properties essential for accurate spatial representation.
This understanding suggests that grid cells are not only crucial for navigation but also
for constructing a precise metric of space, naturally preserving distances and angles. By
elucidating the conditions under which grid cells form a CI map, our work contributes
to the broader knowledge of spatial cognition and may have implications for developing
artificial navigation systems that mimic biological processes.

## Introduction

The brain's ability to navigate relies on specialised spatial cells in the hippocampus and
entorhinal cortex. Among these, entorhinal grid cells are believed to play an important role
[1,2]. Grid cells fire in a characteristic hexagonal pattern across two-dimensional space. They
have been proposed to support path integration, a process by which animals update their
position based on self-motion information—in mechanistic models [3–6], normative models [7–13], and experimental studies [1,14–19]. Additionally, grid cells have been proposed
to form a spatial metric [1,8,20–25], enabling them to represent space in a way that faithfully
encodes distances and directions.

Intriguingly, early normative models [9] produced grid cell-like responses when neural
networks were trained to solve tasks that included path integration. However, the connection
between grid cells and path integration in these models has been shown to be weak [26–28].
Recent advances incorporating joint optimisation for conformally isometric representations—
mappings that preserve angles and scale distances proportionally across space—have demon-
strated improved grid pattern quality [10,12,29], with [11] achieving 100% grid-like cells.
This improvement in grid pattern homogeneity strengthens the connection between the grid
representations and the optimisation objectives they were trained on.

These findings suggest that grid cells might simultaneously serve as the neural substrate for
path integration and as a geometric framework for conformal isometry (CI). However, recent
computational evidence has shown that distance preservation can induce grid patterns, even
without path integration [30]. Thus, whether grid cell patterns are defined by path integration,
distance preservation, or both is still an open question. Since CI leads to improved grid pat-
terns, and grid patterns can emerge without path integration, investigating the CI property of
grid cells in a non-path integrating computational model is important for understanding the
functional role of grid cells for navigation.

To explore the hypothesis that grid cells form a CI more closely, we employ a mechanis-
tic model of grid cells based on a superposition of plane waves [31], characterised by three
free parameters: scale, orientation, and phase. This approach grants us greater control over
the grid cell model, enabling us to directly investigate whether grid cells can support a confor-
mal isometry (CI) and thus serve as a spatial metric. Orientation and scale have been shown
to remain consistent within experimentally recorded grid modules [32], while models sug-
gest that these parameters vary systematically across modules [33,34]. In contrast, grid cell
phases are often assumed to be uniform within modules, as observed in experiments [35] and
implied in models [3]. However, some evidence suggests that the phase distribution may devi-
ate from a purely random uniform arrangement [36]. As phases are the only free parameter
within a module, we investigate whether a single module of grid cells can support a CI and
what phase arrangements are most conducive to achieving this solution.

Our results demonstrate that a module of at least seven grid cells can achieve a near-perfect CI, with optimised phases arranging themselves in a regular hexagonal pattern. Furthermore, we observe that some experimentally recorded grid cell modules exhibit CI-like properties, with one example module displaying a phase arrangement that resembles the hexagonal pattern predicted by our model. This finding provides preliminary support for the hypothesis that grid cells may be organised to support CI, implying that CI properties could be biologically relevant rather than merely a feature of computational models.

Beyond preserving the metric of 2D space, CI-optimised grid cell modules exhibit additional useful properties. We find that CI-optimised modules maintain consistent energy expenditure across space, potentially preventing localised energy spikes that could disrupt function. Furthermore, we explore the minimum number of cells needed to support various spatial encoding tasks, including unique spatial representation, persistent toroidal topology in activity space, and CI formation, all of which are also achieved within a CI-optimised module.

In summary, this study provides both computational insights and preliminary experimental evidence that grid cells may serve as a conformal isometric map of space. By highlighting the role of geometric principles in neural representation, our findings suggest that CI could play a biologically relevant role in grid cell function. Altogether, our findings suggest that grid cells may be inherently suited to form a CI, offering several additional properties that enhance robust and efficient spatial encoding. Further experimental analysis could help to refine and validate these findings.

## Results

### Grid cells as a spatial representation

To model the hexagonal firing patterns of grid cells, we use the well-established approach by [31], wherein idealised grid cell activity is represented as a superposition of three plane waves (As described in Eq 3, and seen in Fig 1a). This setup generates a periodic spatial pattern across two-dimensional space, with the smallest repeating segment, or *unit cell*, forming a hexagonal shape (illustrated in Fig 1b). Within a grid cell module, cells share the same scale and orientation, differing only in phase, which allows the population vector of these cells to exhibit periodicity within each unit cell (as shown in Fig 1c).

The first aspect of our analysis considers how a module of grid cells can uniquely encode spatial positions within its unit cell. Achieving this requires identifying the minimum number of grid cells and phase configurations necessary to create an injective mapping. We find that, theoretically, a configuration of three grid cells can uniquely represent all positions within a unit cell (as proven in S1 Text). This concept is illustrated in Fig 1d, where colours denote the correlation between population activity across spatial locations relative to a reference point (marked by the red cross). Regions with population activity matching the reference point within a tolerance of $\|\underline{g}(\underline{r}) - \underline{g}(\underline{r}')\| < \epsilon = 10^{-2}$ are outlined by white contour lines. For modules of one or two grid cells, ambiguities remain within the unit cell, while a module with three grid cells effectively resolves these ambiguities, enabling a unique spatial representation.

Our second line of inquiry examines the conditions under which the population activity of a grid-cell module forms a toroidal manifold, a concept supported by both theoretical [20,37] and recent experimental studies [38]. Although three grid cells can uniquely represent locations within the unit cell, this configuration is insufficient to establish a toroidal topology. Previous research has suggested that approximately 20 grid cells with randomly arranged phases are required to form a torus [39]. Our observations align with this, as seen in the persistence diagram in Fig 1e. However, we find that a torus can be formed with as few as six cells with an optimal phase arrangement (see Fig C in S1 Text).

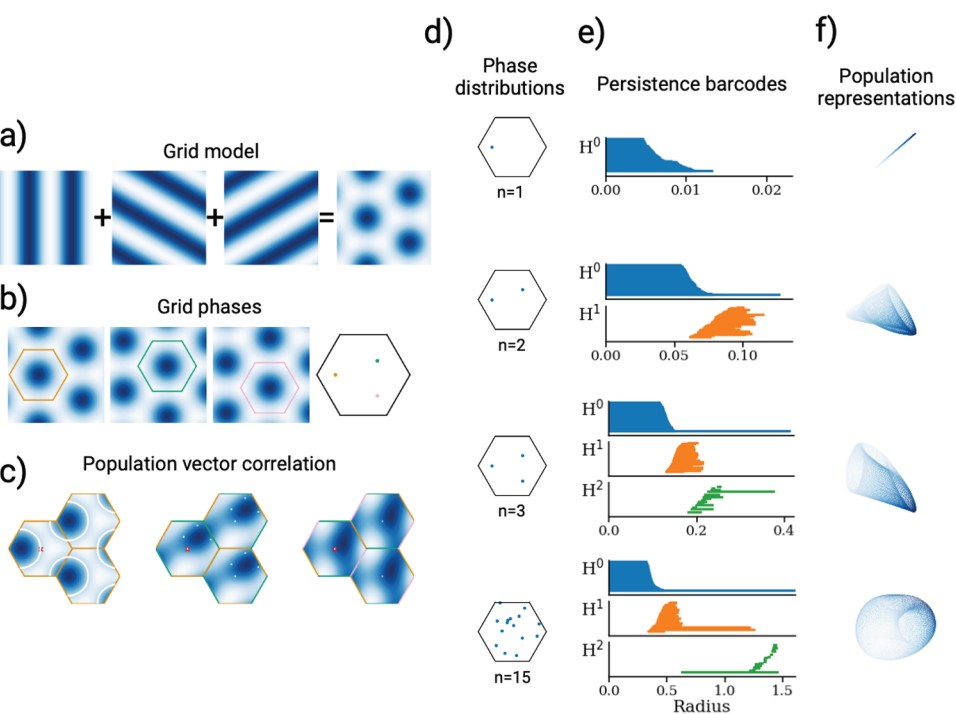

**Fig 1. Unique representation and toroidal encoding in a grid-cell module with varying phases. a)** Illustration of grid cells modelled as a superposition of three plane waves. **b)** Visualization of three distinct grid cells from the same module, each with their respective unit cells superimposed, demonstrating the periodic patterns. The right-hand image shows the phases of the three grid cells within a shared unit cell. **c)** Population vector correlation for 1, 2, and 3 cells' activity (left to right) relative to the activity at the red cross. White lines and dots highlight ambiguous points, defined as locations where the Euclidean distance between the activity vector at each location and that at the red cross is less than $\epsilon = 10^{-2}$. For populations of 1 and 2 cells, ambiguities are present within the unit cell, while with 3 cells these ambiguities are resolved. The unit cell border colours (yellow, green, and pink) correspond to the phases from panel b) included in the population. **d)** Grid modules with varying numbers of cells ($n = 1, 2, 3$, and 15) and randomly distributed phases, shown within a unit cell. **e)** Persistence barcodes of the grid module from d), indicating the lifetime of zero-, one-, and two-dimensional holes in the population representation. With a larger cell number (here $n = 15$), we observe one persistent 0D, two 1D, and one 2D bar, suggesting a toroidal topology. **f)** Population representation of the grid cells from d). The first three plots display the population activity of the grid cells along their firing rate axes. The final plot presents the UMAP projection of the activity of 15 grid cells in three dimensions, resembling a torus. Colours represent the population vector correlation relative to the centre of the unit cell.

## The importance of phase arrangement

As previously discussed, a fundamental attribute of a grid cell module should be its capacity to represent spatial distances uniformly across the environment. This concept aligns with encoding space as a conformal isometry (CI), where spatial relationships are preserved equally in all directions. To illustrate the importance of uniform distance representation, consider a counterfactual scenario: a football field in which spatial scaling is distorted, causing one side of the field to appear disproportionately larger than the other (see Fig 2a). Such an inconsistent spatial metric would make it challenging for players to perceive distances correctly and move efficiently across the field, leading to constant adjustments and confusion. This example highlights why a CI, where spatial distances remain consistent across the environment, is essential for effective navigation and accurate spatial representation.

The metric tensor—a mathematical structure that defines distances and angles on a manifold—serves as a tool to quantify how a representation distorts the spatial variable(s)

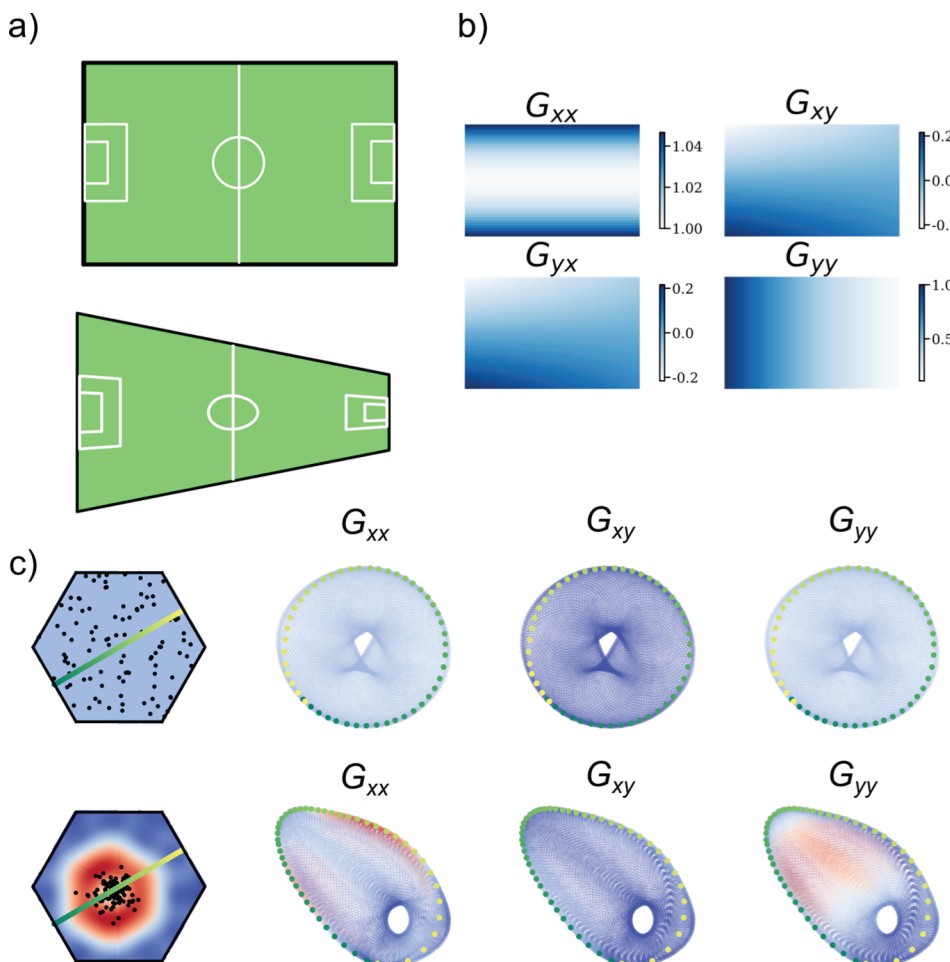

**Fig 2. Distortions in spatial representation are encoded in the metric tensor. a)** An illustration of a standard football field (top) and a distorted version (bottom), where vertical distances compress as one moves to the right. **b)** The metric tensor components show how distances in **a)** are locally deformed in each direction. For example, the $G_{yy}$ component shows that as one goes to the right, distances in the vertical direction will shrink, whereas the $G_{xx}$ component shows that as one goes up or down, distances to the right will expand. **c)** Phase arrangements of 100 grid cells inside a unit cell (upper left in each set of four figures), along with a three-dimensional UMAP projection of the generated activity coloured by the metric tensor components (indicated by labels). Unit cell plots are coloured by the determinant of the metric tensor. The example on the first row shows a low-dimensional projection of a conformally isometric torus, and trajectories are undistorted. The example on the second row shows a projection of the torus, when the phases are sampled densely around a small region. In this case, trajectories crossing phase-dense regions appear more dense on the manifold.

it encodes. Restricting ourselves to the typical experimental setup of an animal navigating a square enclosure (two-dimensional flat Euclidean space), we define the metric tensor as a matrix $G \in \mathbb{R}^{2 \times 2}$ at each point on this two-dimensional manifold. The tensor's components, $G_{xx}$, $G_{yy}$, and $G_{xy} = G_{yx}$ (illustrated in Fig 2b), describe distortions along canonical and joint directions. Uniform representation of spatial distances requires the metric tensor to be diagonal, $G(\underline{r}) = \sigma I$, where $I$ is the identity matrix and $\sigma$ is a scalar.

In the Solstad model, three parameters—scale, orientation, and phase—define each grid cell. However, within a grid-cell module, only the phase varies independently across cells, making it effectively the only free parameter at the module level. For our analysis, we vary

only the phase arrangement to study its impact on spatial representation, as measured by the metric tensor. We begin by examining trajectories on the neural manifold. With an optimal phase configuration that supports a conformal isometry (CI; defined in the following subsection), each trajectory in the environment is proportionally represented within the neural manifold. Conversely, when phases are densely clustered—i.e., not forming a CI—spatial representation expands in phase-dense areas and contracts in sparser regions. These effects are visualised in Fig 2c.

While the main focus of our work concerns idealised grid cells in the open field, biological grid patterns can exhibit distortions in response to e.g. non-symmetric geometries [40]. To investigate whether such distortions can result from grid cells representing non-flat metrics, we optimise representations to learn isometries of a distorted metric. The analysis is presented in "Grid distortions from distorted metrics" in S1 Text, wherein we show that learning CIs in square, flat metric leads to hexagonal patterns, while learning isometries of distorted metrics leads to distorted, elliptical patterns, similar to experimental findings [40] . This analysis was performed using JAX [41,42] and least square fitting of ellipses [43,44].

## Optimal phase arrangements

With an understanding of the metric tensor and how phase arrangement impacts grid cell spatial representation, we explore the minimal number of grid cells required to form a conformal isometric (CI) map and the implications of increasing cell numbers within a module. We also examine phase arrangements in both minimal and large modules.

We begin by optimising the phases of a module containing 1-14 cells for CI (see Fig A in S1 Text), using the loss function defined in Eq 4. We find that there is no appreciable decrease in loss for less than seven cells. For seven cells, however, the loss drops twelve orders of magnitude from $\approx 10^1$ to $\approx 10^{-11}$ during training (optimisation details are described in Training details). This considerable drop in loss indicates that a minimum of seven grid cells is necessary to form a conformal isometry. For 8-14 cells, the loss is also strongly reduced, although this reduction varies with cell count.

To investigate why seven cells are necessary for forming a CI, we observe that the optimised phases in a seven-cell module consistently arrange themselves at the vertices and centre of a regular hexagon (Fig 3a). This arrangement supports both a conformal isometric mapping and a toroidal structure, as evidenced by the UMAP and persistence diagrams (Fig 3b and Fig 3c). The corresponding Voronoi diagram in Fig 3d reveals a partition of space into hexagonal regions, each centred around a phase, suggesting that each cell is optimally positioned to encode distinct portions of the environment, analogous to the efficient coding strategy proposed by [45]. This configuration aligns with principles from convex coding theory, in which each neuron's receptive field corresponds to a specific convex region in stimulus space [46,47]. According to [48], a *good cover* of a toroidal space requires a minimum of seven cells to ensure that overlaps between regions are contractible, preventing disjoint intersections. This criterion also aligns with the map-colour theorem, which specifies that at least seven regions (or "colours") are necessary to cover a torus without two adjacent regions sharing the same designation [49]. We note that this is similar to CI, which requires uniquely encoding displacements in space, but that the map-colour theorem does not require a hexagonal tiling, which is rather caused by CI additionally requiring *equal* encoding of displacements.

Next, we assess whether the hexagonal arrangement is arbitrary or represents a robust solution by conducting grid searches for solution invariances. We vary the orientation and

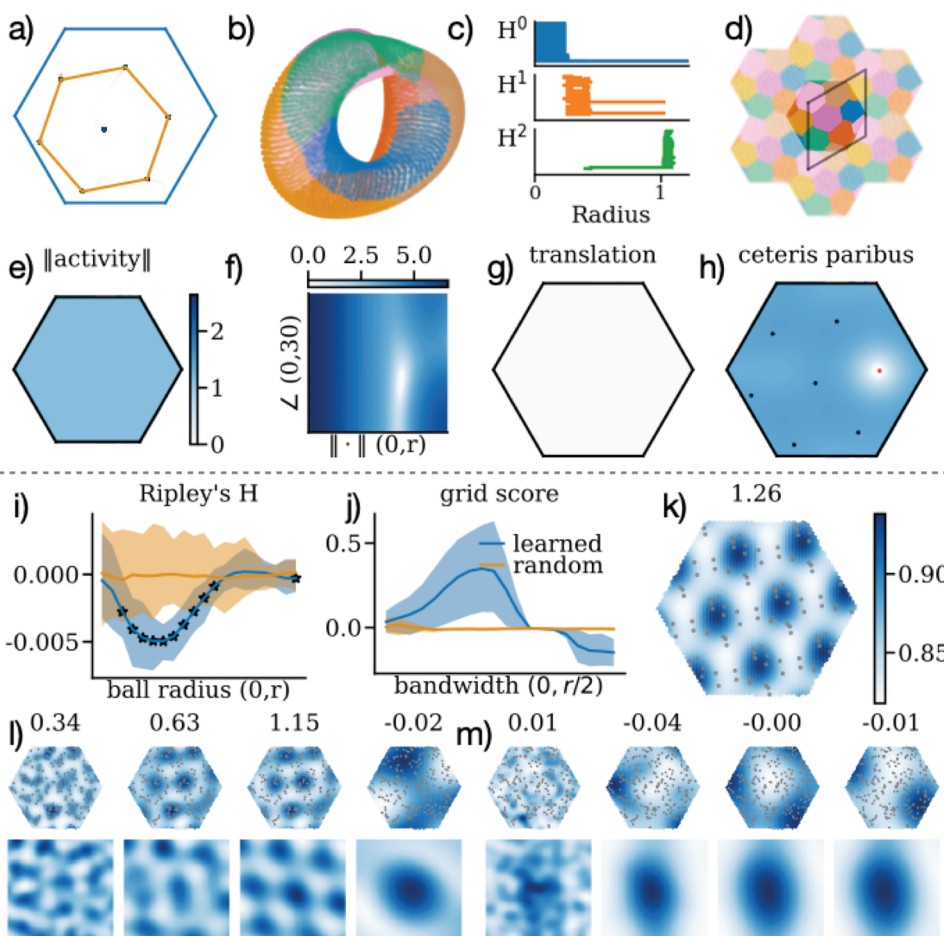

**Fig 3. Optimal phases for seven cells (a-h) and 100 cells (i-m). a)** Learned phase positions for seven grid cells within the unit cell (blue) and their inferred hexagonal arrangement (orange). **b)** Three-dimensional UMAP projection of the seven-cell population activity, colour-coded by Voronoi diagram (shown in d). **c)** A persistence diagram illustrating the activity topology from a). **d)** Voronoi diagram showing spatial partitioning of phases across an extended grid, with the primitive (rhombus) cell superimposed. **e)** L2 norm of population activity across the unit cell. Colour range spans from 0 to $\sqrt{7}$, the maximum norm for seven cells. **f)** Conformal isometry grid search when varying the angle and magnitude of the hexagonal phase solution in a). Colours are $\log \mathcal{C} + 1$ where $\mathcal{C}$ is the conformal isometry metric defined in Eq 5. **g)** Common phase translation grid search, showing invariance to phase shifts (colour scale from f). **h)** Grid search focusing on varying just one (marked as a red dot) of the phases, using the same colour scale as f). **i)** Ripley's H-function analysis comparing the dispersion of optimised (blue) and random (orange) phases. The analysis was conducted on 20 evenly spaced radial distances up to but not including the unit cell radius ($r = 2/3$). Stars indicate significant differences ($p < 0.01$), as determined by a permutation test (details in methods). **j)** Grid score analysis of learned versus random phase distributions, inferred through Gaussian kernel density estimation, plotted against varying kernel bandwidths in the range of (not including) zero to half the radius length of the unit cell $r/2 = 1/3$. **j)** Grid score comparisons of optimised versus random phase distributions across different kernel bandwidths for Gaussian kernel density estimation (KDE). Bandwidths range up to half the unit cell radius ($r/2 = 1/3$), highlighting the hexagonality of the optimised phase arrangement. **k)** Kernel density estimate (KDE) of the 15 replications of the phase solution from panel a), each with an additional random phase shift. Bandwidth for KDE is set at 0.1, with the resulting grid score shown in the title. The KDE is normalized within the unit cell to integrate to unity. **l)** (top) KDE visualisation of optimised phase distributions overlaid with phase locations, with titles indicating grid scores. Four bandwidths were selected: $r/20$, $5r/20$, $10r/20$, and $15r/20$. Standard deviations for the KDEs are $(10^{-1} \cdot 3.17, 10^{-2} \cdot 1.95, 10^{-3} \cdot 2.77, 10^{-6} \cdot 5.93)$, reflecting the KDE colour variability across bandwidths. All KDEs are normalized to have a unit integral. (bottom) Autocorrelograms of the KDEs to confirm hexagonal spatial periodicity. **m)** KDEs for random phase distributions at the same bandwidths as panel l), providing a baseline comparison. KDE standard deviations are $(10^{-1} \cdot 3.81, 10^{-1} \cdot 1.25, 10^{-2} \cdot 4.51, 10^{-3} \cdot 6.50)$.

radius of phases arranged at hexagon vertices and the centre, confirming the solution's robustness to these variations (Fig 3f). We find the optimal radius of the hexagonal phase arrangement (in orange in Fig 3a) to be $0.655 \approx \sqrt{\frac{3}{7}}$ with a 10.9 degree rotation relative to the unit cell radius, consistent with theoretical values (Fig B in S1 Text). Common phase translations also do not alter the loss, indicating translational invariance (see Fig 3g), which we confirm analytically ("Independence of CI solutions" in S1 Text.).

Further, we explore phase interdependencies by moving a single phase while holding others fixed, which shows the collective role of phases in achieving CI (Fig 3h). Analytically, we find that CI solutions are independent, allowing the combination of multiple solutions to form new, valid configurations (see "Independence of CI solutions" in S1 Text.). Computationally, this is verified by generating a module of 15 randomly Tex phase-shifted copies of the hexagonal solution (Fig 3a), with the resulting phases and their kernel density estimates (KDEs) depicted in Fig 3k). The empirical loss scales linearly with the number of copies, further supporting solution independence. Together, these analyses affirm that the seven-cell hexagonal phase arrangement is not simply a product of chance but rather represents an optimal solution. Although the specific arrangement of phases is crucial, the overall positioning of the hexagonal solution is subject to arbitrary phase shifts, suggesting a form of positional invariance. This independence implies a broad landscape of potential CI solutions, potentially allowing combinations of "prime" solutions, like factors of seven cells.

An additional characteristic of CI-optimised grid modules that we find is uniform energy expenditure across space. This implies that the activity norm (L2) remains consistent throughout the unit cell, preventing localized spikes that could disrupt function (Fig 3e). Geometrically, constant energy suggests that the torus resides on an N-sphere, specifically $T^2 \subset S^3 \subset \mathbb{R}^N$, where $N$ is the number of neurons. Thus, CI optimisation simultaneously achieves uniform energy, toroidal topology, and isometric mapping.

Finally, we examine phase distribution patterns for modules with more than seven cells. We trained 50 models with 100 cells from a random uniform initialisation, each for 500 training iterations. The loss begins at approximately $10^0$ and converges to around $10^{-7}$, with a marked decrease by around 180 training steps, indicating efficient learning of CI. Using Ripley's H-function, we compare the regularity of optimised phases to a random uniform distribution [50–52], finding significantly greater dispersal in the CI-optimised phases (Fig 3i). Further, through kernel density estimation with varying bandwidths and grid score analysis, we observe that the optimised phases arrange into a hexagonal pattern (Fig 3j-l). Compared to random configurations (Fig 3m), the CI-optimised phases are distinctly non-random and highly regular, supporting our hypothesis that grid cells are naturally predisposed to achieve a CI through a structured, hexagonal phase distribution.

Additional analysis of modules with random, uniformly distributed phases across a broader range of cell counts further supports the importance of phase organization for CI properties. As shown in Appendix Fig E in S1 Text, simply increasing the number of cells with random phases does not lead to improved CI alignment, reinforcing the conclusion that specific, non-random phase arrangements are crucial for achieving a conformal isometric map.

## Experimental evidence for a CI

To investigate whether grid cells in biological data exhibit properties consistent with a conformal isometry (CI), we analysed an example module of grid cells ($n = 105$) from the

publicly available dataset by [38]. Similar to the approach of [53], we focused on a single module from a single animal (in our case, module 1 from *rat_r_day1*), selected for its high number of grid cells (166 units before thresholding) and extended recording duration in an open field, which provided robust data for analysis. After applying a grid score threshold ($> 0.4$) and retaining only cells with phases contained within the inferred unit cell, 105 cells were included in the final analysis. Our objective was to identify patterns in the inferred metric tensor and phase distribution that align with CI characteristics, as predicted by our computational model. Further details, including ratemaps and grid statistics Fig F in S1 Text, can be found in "Conformal isometry in experimental grid cells" in S1 Text.

We began by examining spatial distances encoded in the neural population vectors. Fig 4a shows the Euclidean distance between the neural population vector at a reference point (marked by the red cross) and vectors at all other spatial locations. On a global scale, the distance map displays hexagonal symmetry and smooth transitions, suggesting that the regularity observed in single grid cells extends to grid cell modules. Locally, we observe isotropy, as distances relative to the reference point appear uniform across directions consistent with conformal isometry.

To further investigate, we computed the metric tensor components for the experimental data (Fig 4b) to assess its approximation to a spatially uniform representation. Using a shared colour scale for all components, we observed that $G_{xx}$ and $G_{yy}$ are similar in magnitude, while $G_{xy}$ is notably lower. This pattern aligns with the properties of conformal isometry (CI), where $G_{xx} = G_{yy}$ and $G_{xy} = 0$ in an ideal case. Additionally, the neural-physical distance plot (Fig 4c) compares distances in the experimental data to configurations with clustered phases or spatially shuffled ratemaps—serving as baselines for non-CI structures. For small physical distances (Fig 4e), we found an approximate linear relationship between neural and physical distances, with low variance in neural distances for small physical distances. This consistency suggests uniform neural encoding of local distances across directions in the environment, supporting an isotropic representation in line with conformal isometry principles.

The one-dimensional histogram of metric tensor values in Fig 4d compares the experimental metric tensor values to those derived from spatially shuffled ratemaps. In the experimental data, the distributions of $G_{xx}$ and $G_{yy}$ are closely aligned, both centred at a non-zero value, while $G_{xy}$ is sharply peaked and centred near zero, consistent with a metric tensor indicative of CI. In contrast, the spatially shuffled ratemaps (perm) display much broader distributions, indicative of a more randomized structure, lacking the sharply defined pattern observed in the experimental data. This comparison further underscores the experimental module's spatial coherence and its alignment with CI properties.

To assess the robustness of CI properties across multiple modules, we calculated the conformal isometry score (CIS) from Eq. (5) for all nine available modules in the dataset, as shown in Fig F in S1 Text. Analysis reveals that while the experimental data aligns more closely with a CI than space-shuffled or phase-clustered configurations, it does not significantly differ from the phase-shuffled baseline ($p = 0.25$, statistic = 43.0). This suggests that while phase arrangements are important for achieving CI properties, the experimental modules in this dataset do not exhibit a markedly stronger CI alignment than randomly rearranged phases. Further tests across additional datasets could provide more comprehensive insights into these findings.

To evaluate phase arrangements, we applied Ripley's H-function (Fig 4e), comparing the observed phase distribution to a random baseline generated from 100 resampling trials, each using the same $32 \times 32$ ratemap resolution and phase inference method as the experimental data. Significant deviations from this baseline, marked by red stars, suggest a degree of structure in the phase distribution that may align with CI properties. Error bars represent

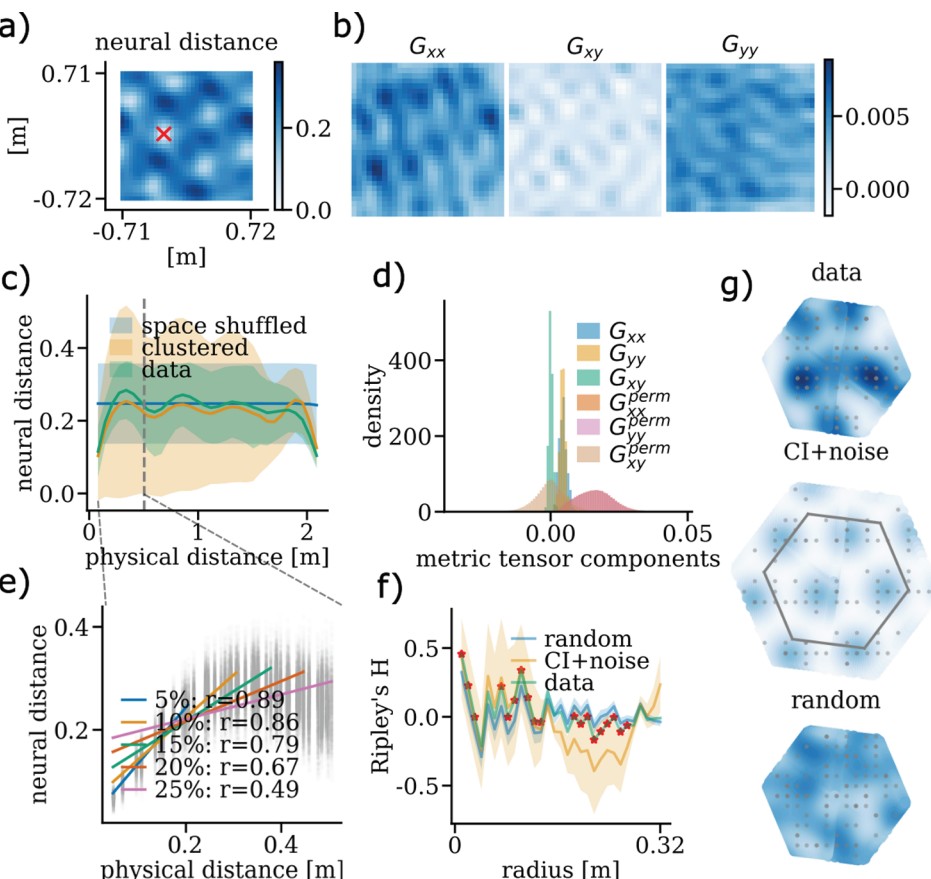

**Fig 4. Signs of conformal mapping and phase hexagonality in a module of 105 experimental grid cells. a)** Euclidean distance between the grid cell population vector at the red cross and population vectors at all other spatial locations. **b)** Metric tensor components for the experimental data, with values represented on a shared colour scale shown in the colour bar. **c)** Neural-physical distance plot comparing experimental data, phase-clustered data, and spatially shuffled neural distances, indicating how neural distances correspond to physical distances. **d)** 1D histogram of the metric tensor components from (b), with distributions from spatially shuffled ratemaps included as a comparative baseline. **e)** Same as (c), but restricted to short physical distances, defined as less than 25% of the maximum physical distance in (c). Linear regression analyses are conducted for subsets corresponding to 5%, 10%, 15%, 20%, and 25% of the total physical distance range. The legend includes the Pearson correlation coefficients (r-values) from the regressions, illustrating the strength of the linear relationship between neural and physical distances for shorter and longer distance ranges. **f)** Ripley's H-function on the experimental phases, with random uniform sampling as a baseline. Shaded areas represent twice the standard deviation from 100 resampling trials, and significant deviations (two standard deviations from the baseline) are marked with red stars. A CI+noise module (105 phases total, derived from 15 noise-induced copies of a 7-cell CI solution) provides an additional comparison, with its radius set at 1.4 times that of the data radius. **g)** Unit cells and kernel density estimates (KDEs) from the grid modules in (e) with phases superimposed. The colour scale is shared across each KDE.

±2 standard deviations from the random baseline distribution. Additionally, we introduce a CI+noise comparison, based on 15 copies of a 7-cell CI solution with added noise sampled from $\mathcal{N}(0, \sqrt{1.4})$. This configuration, derived from ratemaps at the same resolution and using the same phase inference method, provides a reference for interpreting the phase clustering patterns in the experimental data relative to an idealised CI arrangement.

Interestingly, we find that the KDE Ripley's functions align better when the CI+noise module's scale is increased by a factor of 1.4 relative to the scale of the data grid cells. This scaling adjustment may suggest additional spatial features in biological grid cells beyond our model's

CI setup, potentially due to variations in module scaling or measurement noise. Importantly, this is just an example data module; however, the observed deviation from a random phase arrangement suggests that grid phases in this module may lean toward a more structured, dispersed configuration. Further investigation with other datasets is necessary to confirm whether such arrangements are consistent across more grid cell modules.

Finally, Fig 4f presents the kernel density estimates (KDEs) of the grid module's phase distribution, overlaid with the actual phases from the scenarios in e). The KDEs use a shared colour range for comparison.

Together, these analyses suggest that the spatial representation encoded by biological grid cells in this example module may approximate a conformal isometric map. While further investigation with additional data is required to substantiate these preliminary findings, the observed phase distribution and metric properties hint that the neural encoding of space could involve phase configurations consistent with a CI structure, as proposed by our model.

## Benefits of many cells

This section examines the computational advantages of increasing cell count in grid cell modules, focusing on how a larger ensemble enhances encoding robustness, energy efficiency, and spatial resolution. While three grid cells suffice for a bijective transform, six are needed to form a torus, and seven for a conformal isometry (CI). However, modules with more cells offer further computational benefits. Larger ensembles not only mitigate the effects of cell loss but also improve robustness and fidelity in spatial representation. Here, we explore how increasing the number of cells influences the encoding manifold's volume. Specifically, when the model adheres to CI, this volume can be quantified by the *conformal scale*, which impacts both energy consumption and spatial resolution.

To illustrate this, we consider a one-dimensional example where the firing rate of $N$ cells encodes a line interval $r \in [0, 1]$ as:

$$s_i = \cos\left(2\pi r + i\pi/N\right),\qquad(1)$$

where $i$ is the cell index. Fig 5a shows the tuning curves for populations of 2, 3, and 10 cells. Together, these configurations form an isotropic ring within an $N$-dimensional space, as depicted in Fig 5b, establishing a conformal map from the external line to the internal ring representation. The radius of this ring scales with $\sqrt{N/2}$, meaning that as the cell count grows, the ring's circumference expands.

To quantify the change in resolution, consider a population of neurons firing with a discrete instantaneous firing rate (e.g., 0, 1, 2, ..., 200 Hz). This discretisation partitions the ambient space into distinct voxels, each representing a unique state of the population activity. As the radial size of the encoding ring grows, it intersects with an increasing number of these voxels, allowing for finer distinctions between encoded positions, as illustrated in Fig 5b. This increased spatial resolution enables a more precise encoding of the represented interval, thereby enhancing the system's overall fidelity in representing spatial locations.

Second, the expanded ring also results in increased energy consumption. Here, we define "energy" as the total rate of population activity across cells, quantified by the L2 norm of the population activity. With more cells actively contributing to the encoding, the system's overall energy expenditure rises with $\sqrt{N/2}$ (see the larger ring radius in Fig 5b as $N$ increases). This scaling illustrates a trade-off between energy efficiency and encoding precision, where larger cell populations achieve higher resolution at the expense of increased energy.

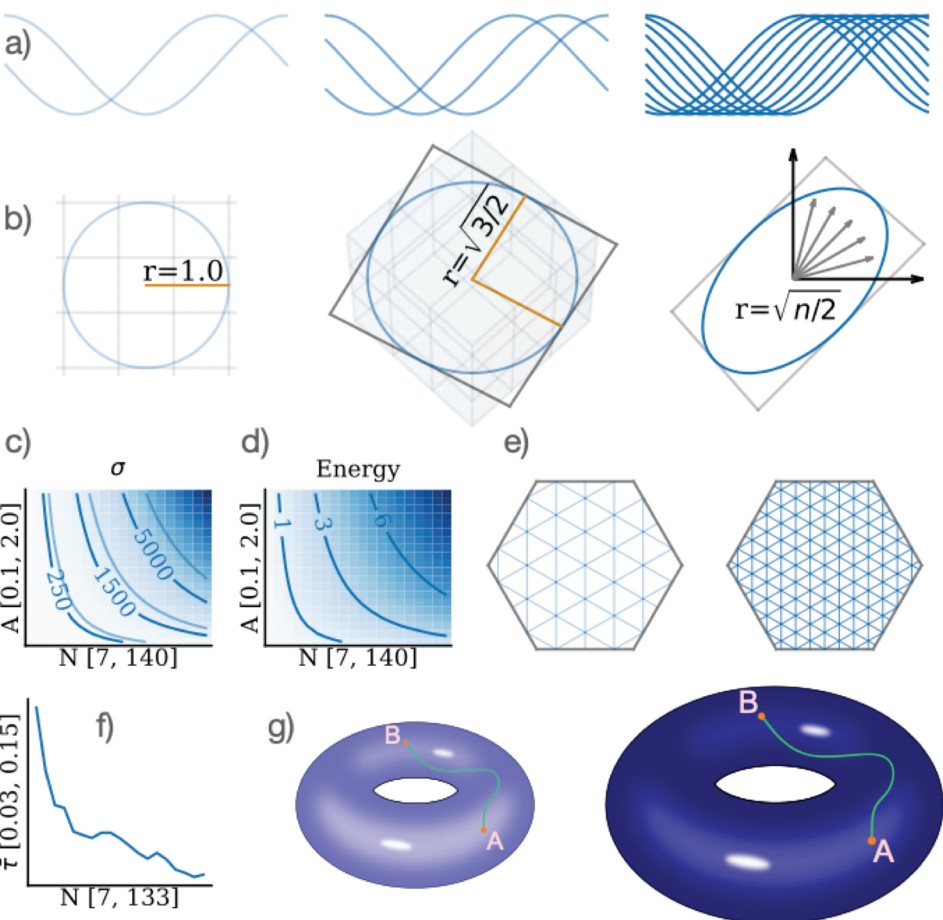

**Fig 5. Impact of the Number of Cells on Spatial Encoding. a)** One dimensional tuning curves for populations of 2, 3, and 10 neurons modelled by Eq 1. **b)** Population activity vectors corresponding to the waves from (a) form an isotropic ring with radius $r = \sqrt{N/2}$, embedded in a plane. Increasing the number of cells expands this ring, intersecting with more unique voxels (distinct population states) and enhancing spatial resolution by enabling finer distinctions between encoded positions. **c)** The conformal scale ($\sigma_{\text{hexagon}}$) from Eq 2 plotted against the firing rate ($A$) and number of cells ($N$), with light blue contour lines representing the square grid's scale ($\sigma_{\text{square}}$). **d)** Population vector norm, or *energy*, shown relative to the firing rate $A$ and cell count $N$, illustrating how energy scales with module size. **e)** A schematic of the unit cell with overlaid meshes of different granularities to depict spatial resolution. The left side, with a coarser mesh, represents a module with fewer cells, while the right side, with a finer mesh, represents a module with more cells, highlighting the resolution enhancement with increased cell count. **f)** The average geodesic distance between initial and optimised phase positions, plotted against module size ($N = 7, 14, 21, \dots, 133$). **g)** Illustration of how increasing cell count leads to larger toroidal structures and extended neural trajectories, highlighting spatial and topological expansion in larger modules.

Finally, increasing the radial size of the ring benefits the signal-to-noise ratio in the encoded variable. Assuming a fixed noise level $\xi$ in the firing rates, the error in the encoded position is bounded by a conformal scaling factor of the error in the neural representation [29]. For a circle of radius $r$, two stimuli $s_1$ and $s_2$ can be distinguished when their distance $d(s_1, s_2) = r\theta$ exceeds $\xi$, where $\theta$ represents the angular separation. Thus, encoding the interval with $N$ neurons allows for the differentiation of stimuli separated by a minimum angle of $\xi/\sqrt{N/2}$ radians. This improvement in signal-to-noise ratio makes the encoded representation increasingly robust as the module size grows.

Importantly, CI-optimised modules achieve consistent signal-to-noise ratios across spatial locations, ensuring uniform representation of space. This contrasts with non-CI-optimised modules, where signal-to-noise ratios vary by location, leaving some regions indistinguishable while others are highly robust. Moreover, for a given spatial resolution or signal-to-noise ratio, CI-optimised networks may require fewer neurons than non-optimised or degenerate configurations, providing a more resource-efficient encoding mechanism, highlighting a potential advantage of a CI-representation.

While the 1D example illustrates how increasing the number of cells enhances spatial resolution and robustness, grid cell modules in biological systems are often considered within a 2D setting—a flat spatial environment with a corresponding 2D toroidal activity manifold. To quantify how these scaling benefits apply to this 2D structure, we derive the conformal scale. Just as in the 1D case where increasing cell count expanded the radius and resolution of the encoding ring, deriving the conformal scale in 2D allows us to assess how the encoding manifold expands with more cells, providing finer spatial resolution and improved robustness.

For a grid cell module with a hexagonal arrangement, the conformal scale, $\sigma_{\text{hexagon}}$, is given by (see derivation in S1 Text):

$$\sigma_{\text{hexagon}} = 3\pi^2 A^2 N. \tag{2}$$

Here, $A$ represents the amplitude of the grid pattern, corresponding to the maximum firing rate of each cell, and $N$ is the number of cells in the module. This solution assumes a conformal isometry and is therefore independent of specific phase arrangements. Furthermore, $\sigma_{\text{hexagon}}$ serves as the scale we optimise in Eq 4.

For comparison, and in line with the analysis in [54], we also derive the conformal scale for a square grid pattern, $\sigma_{\text{square}}$, and find that the ratio $\sigma_{\text{hexagon}}/\sigma_{\text{square}} = 3/2$. This indicates that the hexagonal pattern's scale—and therefore the surface area of the encoding manifold—is 50% larger than that of a module of cells with a square pattern at the same firing rate amplitude, $A$, and cell count, $N$. Consequently, a module with a hexagonal grid pattern exhibits enhanced spatial resolution and improved noise robustness compared to a similar module with a square pattern. Fig 5c visualises this relationship by plotting the conformal scale for the hexagonal pattern, $\sigma_{\text{hexagon}}$, relative to cell count and firing rate, with light blue contour lines providing a reference for the square grid's scale, $\sigma_{\text{square}}$. This comparison highlights the efficiency of the hexagonal firing pattern in grid cells, supporting its role as an optimal configuration for spatial encoding.

Geometrically, increasing the conformal scale of the encoding manifold inflates the manifold, as shown in Fig 5g. This inflation results in several effects already demonstrated in the one-dimensional example: energy (the L2 norm of population activity) increases with cell count (Fig 5d), spatial resolution improves as the manifold intersects more unique population states (Fig 5e), and for a fixed noise level, the encoded signal becomes more prominent, enhancing the signal-to-noise ratio.

Our analyses further reveal that, on average, each phase requires less adjustment from a random uniform initialisation to reach a conformal isometry as the number of cells in the module increases. This finding is illustrated in Fig 5f, where we optimized 19 modules of varying sizes ($N = 7, 14, 21, \ldots, 133$ cells) over 1500 training steps. Each module achieved near-zero loss ($< 10^{-6}$), indicating a successful convergence to a conformal isometric configuration. This reduction in training steps suggests that as module size increases, the solution space of phase configurations compatible with conformal isometry becomes more densely populated, effectively making it easier to achieve CI in larger modules.

### Exploring alternative optimisation objectives

In addition to our primary focus on conformal isometry, we explored two alternative optimisation criteria: (i) minimising linear decoding error and (ii) maximising toroidal persistence. We compared these objectives against a baseline of randomly selected phases as illustrated in Fig 6. For each criterion, we trained 50 models for 5000 epochs, each with 64 uniformly sampled spatial points and a learning rate of 0.001 and cross-evaluated them against the other objectives.

In the case of linear decoding, the model specifically trained to minimise this error initially outperformed the others. However, as the number of cells increased to seven or more, all models converged to similar performance levels. This convergence suggests that beyond a certain point, merely increasing the number of cells does not improve the performance of a linear decoder. This was not the case for a simple non-linear decoder that predicts the $(x, y)$ coordinate by finding the phase of the maximally responsive cell at each point. Moreover, the linear decoding-optimised model generally performed poorly across other criteria, failing to outperform even the random phase baseline. This supports the notion that using a linear readout layer and decoding into Cartesian coordinates appear quite independent of the exact phase distribution. However, since both topological and geometric properties are related to the phases, more expressive non-linear decoders, like cohomological decoding [38,55], are likely to have a stronger dependence on the phases.

The models optimised for toroidal persistence (details in Toroidal loss function) yielded similar CI performance, but achieved a persistent torus for fewer cells compared to a random phase arrangement. On the other hand, models optimised for conformal isometry, not surprisingly, achieved the lowest isometry loss. More surprisingly, however, they also had the most persistent toroidal characteristics for seven cells and upwards, although the gap to random phases and homology-optimised phases decreases with more cells.

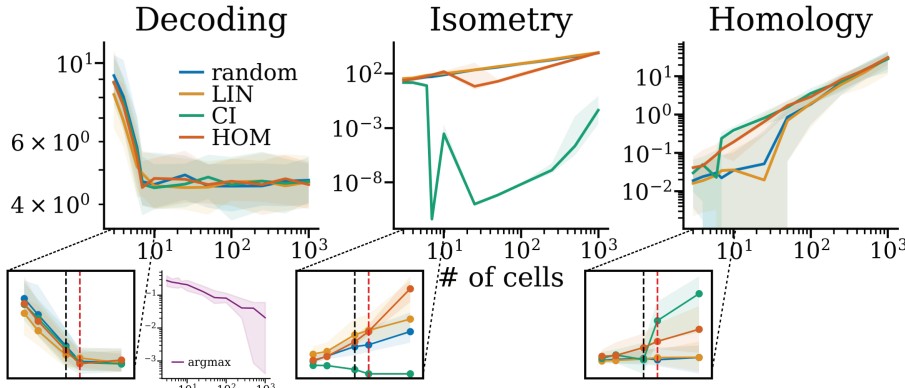

**Fig 6. Model comparison:** Three models, linear decoding (LIN, orange), conformal isometry (CI, green) and homology (HOM, dark orange), along with random phases (blue), evaluated on each other's metrics. In addition, the purple inset shows the decoding error of a non-linear argmax decoder applied to populations with random phases. Zoomed in plots show the performance between 3 and 10 cells in a linear scale, with 6 (black) and 7 (red) cells highlighted by dashed lines.

## Discussion

In this work, we explored the capacity of idealised grid cells with freely varying phases to form a conformal isometry (CI) of two-dimensional flat space. We found that as few as seven cells are sufficient to achieve such spatial encoding. This encoding allows the module to uniquely represent its unit cell while also forming a torus, requiring at least three and six cells, respectively. Remarkably, the phases of these seven cells arrange into a hexagonal lattice—a pattern that persists as the number of cells increases. This hexagonal arrangement enhances spatial resolution and noise robustness throughout the unit cell but comes with increased energy consumption, highlighting a trade-off between precision and energy cost.

Our demonstration that a module of idealised grid cells can form a CI supports the plausibility of this property as fundamental for generating the hexagonal firing pattern, consistent with recent models [10,12,29,34]. This finding reinforces the idea that grid cells provide a neural metric for space [1,8,20–25], preserving the geometric relationships of flat 2D space in the neural representation.

This local metric preservation framework extends to trajectories, ensuring that physical trajectory distances are proportional to their neural counterparts. This property implies that the grid code alone inherits all necessary metric information without requiring additional substrates to account for local distortions (i.e., a neural equivalent of the metric tensor). Consequently, neural trajectory distances computed from the grid code accurately reflects the physical distances travelled. Conversely, if distortions exist and accurate distance computations are desired, another cell ensemble encoding these physical-neural distortions—effectively acting as a metric tensor—would need to be present. Details on trajectory distance calculations, including cases where the grid code does or does not form a CI, are discussed in S1 Text.

In addition to spatial precision and signal-to-noise robustness, we found that a CI maintains constant energy throughout space. This highlights the potential role of energy constraints in grid cell organisation. Specifically, a constant population activity norm across space, akin to the unit normalisation used in [10,12], may facilitate the emergence of hexagonal patterns by minimising energy demands. These findings provide additional insights into why recent normative recurrent neural network models trained for path integration, incorporating CI and energy normalisation constraints, are able to generate robust hexagonal grid patterns.

Beyond the theoretical model, we provide preliminary experimental evidence suggesting that biological grid cells may adhere to CI principles. Analysis of the publicly available dataset from [38] indicates that grid cell modules can exhibit phase arrangements and metric tensor properties consistent with a conformally isometric map. While this evidence is limited to a single module and further replication is necessary, it provides an empirical basis for further testing of CI in biological systems. Large-scale recordings across multiple modules and animals could determine whether this arrangement is a consistent feature of grid cell populations, potentially establishing CI as an organising principle of spatial representation in the brain.

We also investigated the feasibility of encoding spatial positions within a toroidal neural representation and linearly decoding this information from grid cell modules. While random phase arrangements typically require around 20 cells to form a torus consistently [39], our findings show that only six cells are sufficient for accurate toroidal encoding. This positions the minimal set of grid cells for toroidal representation between the number required to

encode a unit cell (3 cells) and the number needed for a CI (7 cells). This raises an open question about the functional significance of toroidal representations and whether they provide computational or biological advantages in grid cell systems.

We observed that compared to a simple argmax decoder, there are significant limitations in linearly decoding Cartesian coordinates from grid cell activity, suggesting that grid cells may encode space in ways incompatible with traditional linear decoding approaches. However, this does not preclude the possibility of linear decoding in alternative coordinate systems, such as polar or manifold-based representations, by downstream cell ensembles. These findings imply that relying solely on linear decoding, as in one of the earliest normative models [7], may fail to capture the true encoding mechanisms of grid cells. Instead, nonlinear decoders, such as cohomological decoding [38,55], may better align with the structure of grid cell representations. These insights highlight the need for more expressive decoding methods in both biological and computational models.

While our model offers valuable theoretical insights, it remains simplified relative to the biological grid cell system and does not capture all experimentally observed phenomena, such as multimodularity [32], pattern distortions [40] and goal-directedness [56]. We have provided preliminary computational evidence that grid distortions can be understood as distortions of the represented metric (see "Grid distortions from distorted metrics" in S1 Text). However, the factors that determine how the represented metric distorts remain unknown.

Furthermore, we have only explored one geometry, so future works could extend this to include environments with other geometries or incorporate salient landmarks, potentially explaining a broader class of observed grid distortions [23,40,56–58]. Future work could also involve distinct grid cell modules, some preserving the geometric properties of physical space, while others distort space based on the animal's internal state or behavioural goals.

How representing a distorted metric impacts navigation performance, is also an unanswered question: In the case of grid cells representing a distorted metric as discussed in S1 Text, accurate computation of traversed distances could require translating between the distorted representation and undistorted representation, possibly necessitating an additional neural ensemble to do so. Additionally, exploring how CI principles apply to multiple interacting modules, path integration, or mapping more complex spaces [59,60] could deepen our understanding of how grid cells flexibly and efficiently encode space.

Our framework primarily focuses on symmetric, non-conjunctive grid cells as the substrate for a conformal isometric (CI) map. Conjunctive grid cells, such as those encoding combinations of spatial and head direction information [61], may also contribute to spatial encoding, but likely through more complex interactions. It is plausible that non-conjunctive grid cells provide the foundational distance-preserving CI representation, while conjunctive grid cells integrate task-relevant variables, such as orientation or behavioural goals. This division of labour suggests a modular organisation within the entorhinal cortex, where distinct grid cell modules specialise in different aspects of spatial cognition. Future research could explore whether conjunctive grid cells are capable of approximating CI properties independently, or whether they rely on non-conjunctive modules to preserve spatial consistency.

Despite its simplicity, our model provides a minimal framework for understanding how grid cells, through phase variation alone, preserve metric properties as a conformal map. Together with preliminary experimental evidence, these findings suggest that CI-based models offer a promising avenue for investigating the functions of grid cells, both theoretically and experimentally.

## Methods

### Grid cell model

We model grid cells following the approach of [31], representing grid cells as a superposition of three plane waves with wave vectors $\underline{k}_j$, each offset by 60 degrees. The model can be expressed as:

$$g_i(\underline{r}) = \frac{1}{3} + \frac{2}{9} \sum_{j=1}^{3} \cos\left(2\pi f R_{60}^{j-1} R_{init} \underline{k}_1 \cdot \left(\underline{r} - \underline{\phi}_i\right)\right), \tag{3}$$

where $i$ indexes grid cells, $R_{init}, R_\theta \in SO(2)$ are rotation matrices determining the grid pattern orientation offset and the relative orientation between wave vectors, respectively. $f$ represents the spatial frequency while $\underline{r} \in \mathbb{R}^2$ and $\underline{\phi}_i \in \mathbb{R}^2$ denote the spatial and phase coordinates, respectively. We set $f = 1$, and by applying the rotation matrix $R_{60}$ we get the three wave vectors, $\underline{k}_1 = (1, 0)^T$, $\underline{k}_2 = R_{60}\underline{k}_1 = (\frac{1}{2}, \frac{\sqrt{3}}{2})^T$ and $\underline{k}_3 = R_{60}\underline{k}_2 = (-\frac{1}{2}, \frac{\sqrt{3}}{2})^T$. The phases are specified in the relevant parts of the text (typically initially random uniform).

### Conformal isometry loss function

To achieve a conformal isometry, we introduce a loss function ensuring that distances in neighbouring points in 2D space are proportional to distances in the grid code. The loss function is defined as:

$$\mathcal{L} = \int_\Omega \left(G_{xx}(\underline{r}) - \sigma\right)^2 + \left(G_{yy}(\underline{r}) - \sigma\right)^2 + 2G_{xy}^2(\underline{r}) d\underline{r}, \tag{4}$$

where, $\sigma$ is the isometry scale, and $G_{xx}$, $G_{yy}$, and $G_{xy}$ are components of the metric tensor $G = J^T J \in \mathbb{R}^{2 \times 2}$. Here, $J_{ij} = J_{g_i}(\underline{r}_j; \Phi) = \frac{\partial g_i}{\partial r_j}$ denotes the Jacobian matrix derived from differentiating Eq 3 with respect to the position $\underline{r}$ for a fixed set of $N$ phases $\Phi \in \mathbb{R}^{N \times 2}$. The notation emphasises the Jacobian's dependency on the phase configuration. The integral over the domain $\Omega$, which represents the unit cell of the grid pattern, is approximated using Monte Carlo integration by uniformly sampling points within the domain.

### Conformal isometry evaluation

To evaluate the degree of conformal isometry achieved by the grid module without explicitly setting the isometry scale we focus instead on the variance and expected values of the metric tensor components over a set of spatial samples. We define the conformal isometry score (CIS) as follows:

$$\text{CIS} = \text{Var}\left(G_{xx}\right) + \text{Var}\left(G_{yy}\right) + \mathbb{E}\left[\left(G_{xx} - G_{yy}\right)^2\right] + 2\mathbb{E}\left[G_{xy}^2\right]. \tag{5}$$

This metric encapsulates the deviation from an ideal conformal isometry, where the variance of $G_{xx}$ and $G_{yy}$ would be minimised, and the expected values of the off-diagonal component $G_{xy}$ and the difference between $G_{xx}$ and $G_{yy}$ would approach zero. A lower CIS value indicates a closer approximation to a true conformal isometry, reflecting a more uniform and isotropic representation of space by the grid module. This provides another way to evaluate the quality of the spatial encoding to form a conformal isometry, without knowing the conformal scale.

## Toroidal loss function

To define a loss function which maximises the topological signatures of a torus, which are two one-dimensional holes and one two-dimensional hole, we followed the approach outlined in [62] and used the `torch_topological` python package [63]. From a grid cell module with randomly initialised phases, we compute the pairwise distances between grid activities and use them to construct a Vietoris-Rips filtration as a function of the scale parameter $\epsilon$. The persistence of a topological feature $\omega$ is defined as the point at which the feature disappears (the death) $d(\omega)$ minus the point at which it first appears (the birth) $b(\omega)$,

$$p(\omega) = d(\omega) - b(\omega). \tag{6}$$

If we then sort all features of dimension $d$ in ascending order according to their persistence $\{p_d(\omega_1), ..., p_d(\omega_N)\}$, we can define our loss function as

$$\mathcal{L}_{hom} = -p_1(\omega_N)^2 - p_1(\omega_{N-1})^2 - p_2(\omega_N)^2 + \sum_{i=1}^{N-2} p_1(\omega_i)^2 + \sum_{j=1}^{N-1} p_2(\omega_j)^2. \tag{7}$$

As one can see, this loss is minimised when the two largest one-dimensional and the largest two-dimensional features are maximised, and all other features disappear.

## Linear Euclidean decoding

To find phases that are optimal for linear decoding we jointly optimised both the phases $\Phi$ and a linear readout matrix $W$ of shape $N \times 2$ for reconstructing a target Cartesian representation of positions $\underline{r} \in \Omega$. The weights $W$ were initialised as $\frac{1}{2N}$ and the phases were initialised randomly uniformly in the unit cell $\Omega$. We start by creating a $T \times 2$ dimensional (response) matrix $P$ of positions sampled uniformly from the unit cell $\Omega$. Subsequently, we construct a corresponding $N \times T$ (design) matrix $X$ of grid cell activities $X_{ij} = g_j(\underline{r}_i)$. The weight matrix $W$ and the phases $\Phi$ were optimised to minimise the loss

$$\mathcal{L}_{\text{Lin}} = \frac{1}{NT} \sum_i^N \sum_j^T d_{hex}((X^T W)_{ij}, P_{ij})$$

where $d_{hex}$ is as described in Shortest distance on hexagonal unit cell.

## Training details

Uniform spatial samples $\underline{r}$ in the unit cell $\Omega$ are generated via rejection sampling using a minimum enclosing square of the unit cell as the proposal distribution. We use the Adam optimiser [64] with default `PyTorch` [65] parameters and mini-batches of size 256 of spatial samples $\underline{r}$, except where specified otherwise. Training length varies between 500 and 10000 training steps for different experiments and is specified throughout the text. We choose training lengths to achieve a small loss, and we observe that models with more cells typically converge faster and, thus, require fewer training steps.

## Low-dimensional projection

We use UMAP [66] to project the activity of multiple grid cells down to three dimensions ($n\_components = 3$). We modify default parameters for Fig 1e by setting $n\_neighbours = 25$.

For Fig 2c we set *n_neighbours* = 40 and *min_dist* = 0.2. For Fig 3b we use *n_neighbours* = 1000.

## Persistent cohomology

All persistence diagrams are computed using the Ripser package [67] with default parameters apart from setting *maxdim* = 2 and *n_perm* = 150.

## Ripley's H-function

We use Ripley's H function [50–52] to analyse phase dispersion or clustering over various scales within a hexagonal domain. It is computed by first wrapping all phases into their unit cell. Then, the phases are duplicated to their six immediately neighbouring hexagons to handle edge effects (see Fig 3d) for an example of the extended tiling). Subsequently, balls with a given radius $\epsilon$ are centred at each (non-duplicated) phase $\underline{\phi}_i$. The core of this analysis involves counting the number of phase points encompassed within each of these balls and normalizing the count. This procedure is mathematically formulated as Ripley's K-function, represented by:

$$K(\Phi, \epsilon) = \frac{|\Omega|}{N(N-1)} \left( \sum_{i=1}^{N} B(\underline{\phi}_i, \epsilon) - N \right) \tag{8}$$

where $B(\cdot)$ signifies the ball counting function (which counts the number of phases within a ball), $N$ is the count of original (non-duplicated) phase points, and $|\Omega|$ denotes the area of the unit cell.

Ripley's K function can be extended to have zero mean, yielding Ripley's H-function, expressed as

$$H(\Phi, \epsilon) = \sqrt{\frac{K(\Phi, \epsilon)}{\pi}} - \epsilon. \tag{9}$$

For the analysis in Fig 3i, we use 100 randomly and uniformly sampled phases within the unit cell as a baseline, and compare them to 100 phases optimised for a conformal isometry.

## Permutation test

In the permutation test, we evaluate whether the response values of two groups, *A* and *B* significantly differ. The test aims to determine the similarity or dissimilarity between these groups using a specific statistic. In our case, we use the sample mean difference, denoted $\bar{x} - \bar{y}$, as the test statistic.

The procedure begins by calculating this statistic between groups *A* and *B*. We then merge the samples from both groups and randomly draw two new samples, $A'$ and $B'$, from this combined pool. The test statistic is recalculated for these new samples. This process of permutation and recalculation is repeated 200 times ($n_{perms}$ = 200), and the resulting statistics are ordered to determine the two-sided percentile (p-value) of the original, non-permuted statistic. The null hypothesis, which posits "no difference" between the groups, is rejected if the p-value is less than the significance level of $\alpha$ = 0.01.

Group *A* consists of Ripley's H scores of 50 CI-optimised models, each with 100 cells. Each model is trained from a random uniform phase initialisation for 500 mini-batches of size 256. The loss for these models and parameters typically decreases from an initial value of around $10^{-1}$ to between $10^{-5}$ and $10^{-8}$. Group *B* comprises 50 models with random uniformly arranged phases within the unit cell, with its sample size matched to that of group *A* ($|B| = |A| = 100$).

The permutation test thus provides a way to evaluate whether the optimised phases in group *A* differ significantly from the random uniform distribution in group *B*, based on Ripley's H statistic.

### Phase KDE

To analyse the distribution of phase points, we apply a kernel density estimate (KDE) using a Gaussian kernel. Initially, we wrap the phases into their unit cell. To capture the long-range periodic effects, we then duplicate the phases to a double-tiling of the unit cell. For the KDE calculation, we utilise the Scipy implementation scipy.stats.gaussian_kde [68]. The bandwidth parameter bw_method is detailed in the description of each analysis.

### Grid score

To assess the hexagonal symmetry in the phase distributions, we employ a modified grid score computation. The process begins with the calculation of the Gaussian phase KDE, as detailed in Phase KDE. This KDE is then evaluated over two square mesh grids: a smaller grid covering the interval $[-r, r]^2$ in $\mathbb{R}^2$ with a resolution of $64 \times 64$ pixels, and a larger grid spanning $[-3r/2, 3r/2]^2$ with $127 \times 127$ pixels. These grids, or ratemaps, are spatially correlated using the Pearson correlation coefficient in 'valid' mode, producing an autocorrelogram of $64 \times 64$ pixels. This autocorrelogram corresponds to the minimal enclosing square of the unit cell, maintaining the periodicity of the pattern.

Next, we construct a binary annulus mask matching the autocorrelogram's dimensions. The outer circle of the annulus is defined by the unit cell's radius, while the inner circle's width is set equal to the KDE kernel bandwidth, as specified in the text. This masked autocorrelogram is then correlated with rotated versions of itself at angles of $30, 60, 90, 120, 150,$ and $180$ degrees, again using the Pearson correlation coefficient.

The final grid score is derived by calculating the mean difference between the correlation coefficients at angles of $60, 120, 180$ degrees and those at $30, 90$ and $150$ degrees. This method effectively quantifies the degree of hexagonal symmetry present in the phase distributions.

### Hexagonal unit cell mesh

To create a hexagonal mesh, we start by generating a square mesh of spatial points $\underline{r}$ within the interval $[0, 3R/2]^2$, where $R$ is the radius of the unit cell. This mesh is initially conceptualised in a 60-degree rhombus basis relative to the standard basis. We invert this representation to the standard basis with the inverse rhombus transform $T^{-1}$. The basis change matrix is, thus, given by $T = (\underline{e}_1, R_{60}\underline{e}_1)^T$ where $\underline{e}_1 = (1, 0)^T$ and $R_{60}$ a 60-degree (counter-clock wise) rotation matrix. Finally, we wrap the corresponding rhombic coordinates to the hexagonal unit cell.

### Shortest distance on hexagonal unit cell

To determine the shortest distance between two points, $\underline{r}_1$ and $\underline{r}_2$, within a hexagonal unit cell with periodic boundaries, we select one point $\underline{r}_2$ and replicate it in all nearest neighbour unit cells. The shortest distance respecting periodic boundary conditions is acquired by selecting the shortest out of all distances between $\underline{r}_1$ and $\underline{r}_2$ including the six replicas.

### Voronoi diagram construction

We construct a Voronoi diagram, categorising each spatial point of a Hexagonal unit cell mesh. Each point is categorised to belong to the phase that it has the Shortest distance on hexagonal unit cell to.

## Calculation of learning trajectory length

To quantify how the phases evolve during training, we calculate the learning trajectory length. This metric represents the average distance travelled by the phases from their initial positions $\underline{\phi}^0$ to their final positions at the end of training $\underline{\phi}^T$. We define the learning trajectory length $\bar{\tau}$ as:

$$\bar{\tau} = \frac{1}{N} \sum_{i=1}^{N} d(\phi_i^0, \phi_i^T). \tag{10}$$

Here, $N$ is the number of phases, and $d(\cdot)$ represents the distance function as defined in Shortest distance on hexagonal unit cell. By averaging these distances across all phases, $\bar{\tau}$ effectively captures the overall extent of movement or adjustment the phases undergo during the learning process.

## Experimental ratemaps

To test the conformal isometry (CI) hypothesis, we utilise the publicly available dataset from [38]. The experimental CI evidence presented in Fig 4 is based on ratemaps from *rat_r_day1*, module 1. These ratemaps are shown alongside the module's corresponding grid statistics in Fig F in S1 Text.

Ratemaps are generated using `scipy.stats.binned_statistic_2d` with 32 bins. Missing or unvisited locations are filled, and the data is smoothed using `astropy.convolution.convolve` with wrapping and a Gaussian kernel (`astropy.convolution.Gaussian2DKernel`) with a width parameter of 2.

Grid statistics, including local peak locations, grid score, peak-to-peak spacing, and orientation of the grid pattern, are inferred from the ratemaps using the `find_peaks`, `gridness`, and `spacing_and_orientation` functions provided by the spatial-maps library.

The dataset for *rat_r_day1*, module 1, includes 166 cells, of which we analyse 105. Cells are excluded based on two criteria: (1) grid scores below 0.4 and (2) phases that fall outside the unit cell defined by the inferred orientation and peak-to-peak distance of the grid pattern.

## Experimental neural distance

Neural distance is calculated as the population vector norm of the grid module activity, $\|\underline{g}_{\text{data}}(\underline{r}') - \underline{g}_{\text{data}}(\underline{r})\|_2$, between all ratemap bin positions $\underline{r}$ and a reference point $\underline{r}'$. In Fig 4a, the reference point is marked with a red X. In Fig 4b, all-to-all distances are computed using all reference locations.

## Experimental metric tensor

To compute the metric tensor components from the experimental data, we first calculate the gradients of the ratemaps $\Delta\mathcal{R}$ for each grid cell (see Experimental ratemaps for a description of how to calculate ratemaps). Gradients are obtained using a second-order finite difference approximation, as implemented in `numpy.gradient`. The resulting gradient ratemap tensor has dimensions $\dim(\Delta\mathcal{R}) = n \times 32 \times 32 \times 2$, where $n$ is the number of grid cells, and the last dimension corresponds to the gradient of the two spatial directions.

The metric tensor components $G_{xx}(\underline{r})$, $G_{xy}(\underline{r}) = G_{yx}(\underline{r})$, and $G_{yy}(\underline{r})$, evaluated at spatial positions $\underline{r}$ defined by the ratemap bins, are computed as population vector inner products $\langle \cdot, \cdot \rangle$ of the ratemap gradients along the respective spatial directions:

$$G_{xx} = \langle \Delta_x \mathcal{R}, \Delta_x \mathcal{R} \rangle, \quad G_{xy} = G_{yx} = \langle \Delta_x \mathcal{R}, \Delta_y \mathcal{R} \rangle, \quad G_{yy} = \langle \Delta_y \mathcal{R}, \Delta_y \mathcal{R} \rangle.$$

For brevity, we have omitted the explicit dependence on spatial coordinates $\underline{r}$. These metric tensor components capture local spatial relationships within the grid cell population, enabling an analysis of how well the experimental data aligns with conformal isometry properties.

## Supporting information

**S1 Text Supporting information text.**
(PDF)

**Fig A in S1 Text Conformal isometry loss history.** for 1–7 cells (left panel) and 8–14 cells (right panel). The numbers in parentheses in the legend indicate the cell count for the right panel. The y-axis range for the right panel is identical to that of the left panel.
(PDF)

**Fig B in S1 Text Visualisation of the 7-colour map of a flat hexagonal torus. a, b)** 7 hexagons (coloured/grey), with side lengths $a$, optimally and recursively arranged to tile a plane, are contained in repeating, larger hexagons (black) of radius $R = \sqrt{7}a$ or rhombi (white/red) of side lengths $A = \sqrt{21}a$, tilted $\phi = \arctan\frac{\frac{\sqrt{3}}{2}a}{\frac{\sqrt{9}}{2}a} \approx 10.89$ degrees compared to the orientation of the smaller hexagons.
(PDF)

**Fig C in S1 Text Persistence plot of the largest one dimensional feature (blue), the second largest one dimensional feature (orange) and the largest two dimensional feature (green) in a CI models with different numbers of cells.**
(PDF)

**Fig D in S1 Text Sanity check for Ripley's H function.** The left panel shows 15 copies of a module of 7 cells optimised for CI with varying degrees of common normal noise, as indicated by the legend. The dashed line at $2/f\sqrt{21}$ represents the expected distance between phases arranged as hexagons with size 1/7 of the unit cell. The right panel provides a baseline of random uniform phases, along with 15 copies of the 7-cell module optimised for CI with uniformly distributed random noise shifts.
(PDF)

**Fig E in S1 Text Conformal Isometry Score (CIS) Does Not Improve with More Cells in a Module with Random Phases.** Plot of the conformal isometry score (CIS) normalized by the cell count (Ncells) for modules of varying sizes with random, uniform phase arrangements. Results suggest that increasing the number of cells alone does not lead to an emergent conformal isometry.
(PDF)

**Fig F in S1 Text 105 Experimental Grid Cell Ratemaps from the publicly available dataset by [38].** The cells are from *rat_r_day1*, module 1. **a)** Grid statistics for the module, with the red line indicating inferred spacing and orientation used to set the unit cell parameters for the data. **b)** Ratemaps with a common colour range, indicated by the colour bar. The unit cell is superimposed, with inferred phases marked by red dots.
(PDF)

**Fig G in S1 Text Violin Plot of Conformal Isometry Scores for Experimental and Baseline Modules for the nine modules in the publicly available dataset by [38].** The "Phase Shuffled" condition applies random shifts in the x and y directions to the ratemaps, while the

"Phase Clustered" condition centres all phases. "Space Shuffled" randomly redistributes the spatial coordinates of the ratemaps.
(PDF)

**Fig H in S1 Text  Verifying the conformal scaling law. a)** Learned conformal scale ($\sigma$) as a function of the number of grid cells ($n$) in the module. Also indicated is the theoretically predicted scale (dashed line). **b)** Metric components for a module of $n = 100$ grid cells with trainable phase distribution and scale.
(PDF)

**Fig I in S1 Text  Grid distortions, geometries and non-flat metrics. a)** Samples used for training models, drawn uniformly in the explored arenas **b)** The distorted metric to be learned by the model. **c)** Loss history for a baseline square environment with the flat metric (FS), a trapezoid with the flat metric (FT), and a trapezoid arena with the distorted metric in b) (DT). d) Distribution of unit pattern eccentricity, for the different models. **e)** Learned representations (left) and corresponding induced metrics (right) for each model.
(PDF)

## Acknowledgments

We want to thank Adrian Kirkeby for an insightful discussion.

## Author contributions

**Conceptualization:** Vemund Sigmundson Schøyen, Kosio Beshkov, Markus Borud Pettersen, Mikkel Elle Lepperød.

**Data curation:** Vemund Sigmundson Schøyen.

**Formal analysis:** Vemund Sigmundson Schøyen, Kosio Beshkov, Markus Borud Pettersen, Erik Hermansen, Konstantin Holzhausen.

y**Funding acquisition:** Anders Malthe-Sørenssen, Marianne Fyhn.

**Investigation:** Vemund Sigmundson Schøyen, Kosio Beshkov, Markus Borud Pettersen, Erik Hermansen, Konstantin Holzhausen.

y**Methodology:** Vemund Sigmundson Schøyen, Kosio Beshkov, Markus Borud Pettersen, Erik Hermansen, Konstantin Holzhausen, Anders Malthe-Sørenssen, Mikkel Elle Lepperød.

**Project administration:** Vemund Sigmundson Schøyen, Mikkel Elle Lepperød.

**Resources:** Anders Malthe-Sørenssen, Marianne Fyhn.

**Software:** Vemund Sigmundson Schøyen, Kosio Beshkov, Markus Borud Pettersen, Erik Hermansen, Konstantin Holzhausen.

**Supervision:** Vemund Sigmundson Schøyen, Anders Malthe-Sørenssen, Mikkel Elle Lepperød.

**Validation:** Vemund Sigmundson Schøyen, Kosio Beshkov, Markus Borud Pettersen, Erik Hermansen, Konstantin Holzhausen.

**Visualization:** Vemund Sigmundson Schøyen, Kosio Beshkov, Markus Borud Pettersen, Erik Hermansen.

**Writing – original draft:** Vemund Sigmundson Schøyen, Kosio Beshkov, Markus Borud Pettersen, Erik Hermansen, Konstantin Holzhausen, Anders Malthe-Sørenssen, Marianne Fyhn, Mikkel Elle Lepperød.

**Writing – review & editing:** Vemund Sigmundson Schøyen, Kosio Beshkov, Markus Borud Pettersen, Erik Hermansen, Mikkel Elle Lepperød.

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
