## [Decision Letter · Decision Letter 0]

14 Aug 2024

Dear Dr. Lepperød,

Thank you very much for submitting your manuscript "Hexagons all the way down: Grid cells as a conformal isometric map of space" for consideration at PLOS Computational Biology.

As with all papers reviewed by the journal, your manuscript was reviewed by members of the editorial board and by several independent reviewers. In light of the reviews (below this email), we would like to invite the resubmission of a significantly-revised version that takes into account the reviewers' comments.

We cannot make any decision about publication until we have seen the revised manuscript and your response to the reviewers' comments. Your revised manuscript is also likely to be sent to reviewers for further evaluation.

Sincerely,

Cristina Savin, Ph.D

Guest Editor

PLOS Computational Biology

Lyle Graham

Section Editor

PLOS Computational Biology

Reviewer's Responses to Questions

**Comments to the Authors:**

Reviewer #1: The authors present an analysis of grid cells and their numbers from the perspective of injective mappings, conformal isometry, energetic considerations and attractor topology and geometry. In particular, they consider a plane-wave-based model for grid cells, and find (through computational and analytical results) the minimum number of grid cells and their particular relative phases necessary to form a conformal isometry of the traversed flat space within the neural representational space. Furthermore, they also establish that achieving such conformal isometry becomes easier with increasing number of cells in a given module.

Through their results, the authors have presented a novel approach to understanding the geometry of the neural encoding of grid cells. As such, I think their work will certainly be of strong interest to the mathematically inclined community that is interested in neural representations and grid cells. However, I have a small number of questions/comments that I feel need to be addressed before recommending their work for publication:

1) The motivation in the early introduction seems misplaced/: The authors state, "These findings indicate that grid patterns can emerge without being directly linked to path integration, raising questions to their exact function in the brain." This is not necessarily true, or at least does not follow from the findings described. Specifically (as the authors correctly note) the findings in Refs. [13-18] demonstrate path integration objectives can be achieved without the emergence of hexagonal-like grid cells. This only indicates that path integration is not directly linked to grid patterns. These results do not show grid patterns emerging without being related to path integration. Perhaps the authors intended to state that these findings indicate that path integration can be achieved without grid patterns, raising questions to the necessity and importance of the grid patterns?

2) What is being depicted in Fig. 1d? While I follow the gist of what is being demonstrated, it is unclear what is being plotted to show the population vector correlation, and how the set of ambiguous points is being determined. Some additional clarification in the caption or main text would make it easier to follow

3) The authors present an interesting claim relating the minimum number of cells necessary to obtain a conformal isometry, and the minimum number of colors needed for a planar graph on the torus. They relate this through the cells arranged as the vertices and center of a regular hexagon, and the corresponding regular Voronoi tessellation being 7-colorable. However, any 2d periodic lattice of activity would lead to a torus (for example, a square lattice instead of a hexagonal lattice). From the torus and it’s coloring, one continues to require 7 colors, but from the perspective of achieving a CI the cells may not appear as lying on a hexagon and its center? How does one relate a general result of coloring on a torus to a specific form of achieving the torus through hexagonal patterning (rather than any other shape)? Some clarification in this regard would certainly be helpful.

4) A lot of the discussion in Sec. 2.5 is around the number of cells being around 6-7. However, the figure corresponding to this section (Fig. 5) is presented on a logarithmic scale and is a little hard to parse exactly what is qualitatively changing around 6-7 cells. Perhaps a zoom in around the smaller number of cells could be included as an inset of an appendix figure?

5) In Fig 5 right, why does the curve corresponding to training on the homology loss/toroidal persistence not perform the best based on the homology metric?

6) In the introduction, the authors state that they demonstrate that "Cartesian coordinates cannot be perfectly linearly decoded from a grid cell module". Similarly, in the discussion, they state "we found that the linear decoding of Cartesian coordinates from grid cell activity faces significant limitations". This does not seem to be stated or implied in the results section discussing linear decoding. Is the idea here that 6 grid cells are sufficient for a torus, but that linear decoding loss does not decrease until 7 grid cells and above? This seems like a weak claim, since for larger than 6 cells the decoding loss seems to be quite small across the types of models (cf. Fig. 5). The authors should clarify what is meant here.

7) The legend for Fig A1 seems somewhat cryptic. Do the numbers in the brackets indicate the curves in the right panel? Also, does the right panel have the same scale of the y-axis as the left panel? If so, why do some larger numbers of cells, such as the green curve (10 cells?) have such significantly higher losses?

Reviewer #2: The review is uploaded as attachement

Reviewer #3: The manuscript by Schoyen and colleagues explores the hypothesis that grid cells in the brain form a conformal isometric map, i.e., they preserve angles and distances while mapping the external world. Using a relatively simple framework and model, they demonstrate that at least seven grid cells with phases arranged in a hexagonal pattern are necessary to achieve a conformal isometry. Increasing the number of grid cells improves spatial resolution but also increases energy consumption. They also comment on the feasibility of 2D decoding of spatial position from grid cells.

The manuscript is overall interesting and well-written. The concepts and intuition have been well introduced. Overall, I have some major conceptual questions and suggestions for the authors.

1. While the authors have commented and referenced the effect of environmental boundaries in distorting the grid (page 15), this experimental result has not been sufficiently discussed and modeled in the light of their results.

Several studies showing grid cells in trapezoids (Krupic et al, 2014), 3D space (Ginosaur et al, 2014), 1D space (Domnisoru et al, 2013) do not have regular hexagonal geometry, but are rather distorted. Moreover, grid cells respond in various ways when animals are performing task or when features of the environment are modified (Peng et al, 2023, biorxiv, Diehl et al, 2016, Boccara et al, 2019 and many more), modulating their rates as well as translocating towards goals.

The vast majority of experimental studies demonstrate that under regular conditions of navigation, grid cells do not form a conformal isometric map. The authors must address this discrepancy between their predictions and observed experimental results.

2. Similarly, the authors should comment on how this framework applies to conjunctive grid cells in the entorhinal cortex. Do they propose that only the select, non-conjunctive grid cells (those forming the toroid as reported in Gardner et al, 2022) are the ones that form the CI? More broadly, how do they envision this network integrating within the circuits in the entorhinal cortex?

3. The 2D decoding results are interesting, but somewhat underexplored. “Furthermore, we found that the linear decoding of Cartesian coordinates from grid cell activity faces significant limitations, suggesting a potential mismatch between traditional linear decoding models and the actual grid cell encoding mechanisms.” Can the authors expand on this? Perhaps I misunderstood, but if the grid cell network preserves distances, wouldn’t one expect that linear decoding would be effective for estimating Cartesian coordinates? Are there alternative decoding approaches that the authors can recommend?

4. I have some objection to their distinction between grid cells performing path integration and grid cells as a metric for space. The entire manuscript is written as if these are contrasting viewpoints, while these are quite complementary. Perhaps the authors can clarify the major differences in their view between these concepts?

**Have the authors made all data and (if applicable) computational code underlying the findings in their manuscript fully available?**

Reviewer #1: Yes

Reviewer #2: Yes

Reviewer #3: Yes

PLOS authors have the option to publish the peer review history of their article (what does this mean?). If published, this will include your full peer review and any attached files.

Reviewer #1: No

Reviewer #2: No

Reviewer #3: No
---

## [Decision Letter · Decision Letter 1]

29 Dec 2024

PCOMPBIOL-D-24-00824R1

Hexagons all the way down: Grid cells as a conformal isometric map of space

PLOS Computational Biology

Dear Dr. Lepperød,

Thank you for submitting your manuscript to PLOS Computational Biology. After careful consideration, we feel that it has merit but does not fully meet PLOS Computational Biology's publication criteria as it currently stands. Therefore, we invite you to submit a revised version of the manuscript that addresses the points raised during the review process.

We also ask that you briefly address the decoding question of Reviewer 1, and take into account the suggestions of Reviewer 2 vis-a-vis the presentation.

Please submit your revised manuscript within 30 days Feb 28 2025 11:59PM. If you will need more time than this to complete your revisions, please reply to this message or contact the journal office at ploscompbiol@plos.org. Please include the following items when submitting your revised manuscript:

We look forward to receiving your revised manuscript.

Kind regards,

Cristina Savin, Ph.D

Guest Editor

PLOS Computational Biology

Lyle Graham

Section Editor

PLOS Computational Biology

**Journal Requirements:**

1) Please upload the figures in the online submission form in a correct numerical order from (1-6).

2) Please ensure that the affiliations of the authors listed on the manuscript title page do exactly match with the affiliations provided in the online submission form.

NOTE: Affiliations should include a department (if applicable), an institution, a city, and a country

4) Please provide a complete Data Availability Statement in the online submission form.

**Reviewers' comments:**

Reviewer's Responses to Questions

**Comments to the Authors:**

**Please note that the review is uploaded as an attachment.**

Reviewer #1: I think the authors have sufficiently responded to almost all of my questions and comments. In its current form, I think the paper will be well appreciated by the mathematical community interested in grid cells, and will be an appropriate fit at PLOS Comp Bio. I do however have a one minor comment, relating to a previous question that I had raised:

I am still unclear on the claim being made in relation to linear decoding of Cartesian coordinates from grid cell activity. Since the linear decoding loss is quite small for anything more than 6–7 cells, I do not know if it is fair to state "We observed significant limitations in linearly decoding Cartesian coordinates from grid cell activity, suggesting that grid cells may encode space in ways incompatible with traditional linear decoding approaches". I do not follow why the authors believe that there is some fundamental incompatibility --- perhaps some additional clarification could help.

Reviewer #2: The authors have answered satisfactorily all of our concerns. We have some minor suggestions that we think would improve clarity, but leave it to the authors to decide whether or not to include them.

1 - We suggest that the authors revise the color coding for the tensor metric plots to ensure that the colors used for these measurements do not overlap with the colors used for different types of optimizations. Several axis labels lack units (for example ‘physical distance’ in Fig. 4d), and some axes lack labels.

2 – Figure 1: Including a color bar that indicates the minimum and maximum correlation values directly in the figure, instead of describing the color scheme in the caption, would make the figure more self-explanatory and easier to interpret.

3 – Figure 4. The new Figure adds new depth to the article, but the plots could be clearer. In panel d we suggest finding a way to avoid the overlap between histograms, as some of them are hidden in the background. In panel c we suggest finding a way to focus on the linear part supporting the claims. We also suggest clarifying what is meant by "spatially shuffled" in the figure caption.

4 - The sentence “Notably, we find that for small physical distances, there is a…” could benefit from clarification. Does "small physical distances" refer to distances within the spacing of grid cells?

Reviewer #3: I find the updated manuscript greatly improved, with the additions in the discussion and analysis expanding the scope of the manuscript to a broader audience. The authors have addressed my comments and I have no further concerns.

**Have the authors made all data and (if applicable) computational code underlying the findings in their manuscript fully available?**

Reviewer #1: Yes

Reviewer #2: Yes

Reviewer #3: Yes

PLOS authors have the option to publish the peer review history of their article (what does this mean?). If published, this will include your full peer review and any attached files.

Reviewer #1: No

Reviewer #2: No

Reviewer #3: No

**Figure resubmission:**
---

## [Editor Report · Decision Letter 2]

16 Jan 2025

Dear Dr. Lepperød,

We are pleased to inform you that your manuscript 'Hexagons all the way down: Grid cells as a conformal isometric map of space' has been provisionally accepted for publication in PLOS Computational Biology.

Best regards,

Cristina Savin, Ph.D

Guest Editor

PLOS Computational Biology

Lyle Graham

Section Editor

PLOS Computational Biology

---

## [Editor Report · Acceptance letter]

PCOMPBIOL-D-24-00824R2

Hexagons all the way down: Grid cells as a conformal isometric map of space

Dear Dr Lepperød,

I am pleased to inform you that your manuscript has been formally accepted for publication in PLOS Computational Biology. Your manuscript is now with our production department and you will be notified of the publication date in due course.

With kind regards,

Zsofia Freund
